# Intradialytic Nutrition and Hemodialysis Prescriptions: A Personalized Stepwise Approach

**DOI:** 10.3390/nu12030785

**Published:** 2020-03-16

**Authors:** Giorgina Barbara Piccoli, Francoise Lippi, Antioco Fois, Lurlynis Gendrot, Louise Nielsen, Jerome Vigreux, Antoine Chatrenet, Claudia D’Alessandro, Gianfranca Cabiddu, Adamasco Cupisti

**Affiliations:** 1Dipartimento di Scienze Cliniche e Biologiche, Università di Torino, 10100 Torino, Italy; 2Centre Hospitalier Le Mans, 72037 Le Mans, France; flippi@ch-lemans.fr (F.L.); jvigreux@ch-lemans.fr (J.V.); achatrenet@ch-lemans.fr (A.C.); 3Nephrologie, Centre Hospitalier Le Mans, 72000 Le Mans, France; afois@ch-lemans.fr (A.F.); lgendrot@ch-lemans.fr (L.G.); lnielsen@ch-lemans.fr (L.N.); 4Department of Clinical and Experimental Medicine, University of Pisa, 56126 Pisa, Italy; dalessandroclaudia@gmail.com (C.D.); adamasco.cupisti@med.unipi.it (A.C.); 5Nephrology, Brotzu Hospital, 09100 Cagliari, Italy; gianfranca.cabiddu@tin.it

**Keywords:** hemodialysis, hemodiafiltration, albumin, Kt/V, malnutrition, elderly, comorbidity, MIS index, dialysis efficiency

## Abstract

Dialysis and nutrition are two sides of the same coin—dialysis depurates metabolic waste that is typically produced by food intake. Hence, dietetic restrictions are commonly imposed in order to limit potassium and phosphate and avoid fluid overload. Conversely, malnutrition is a major challenge and, albeit to differing degrees, all nutritional markers are associated with survival. Dialysis-related malnutrition has a multifactorial origin related to uremic syndrome and comorbidities but also to dialysis treatment. Both an insufficient dialysis dose and excessive removal are contributing factors. It is thus not surprising that dialysis alone, without proper nutritional management, often fails to be effective in combatting malnutrition. While composite indexes can be used to identify patients with poor prognosis, none is fully satisfactory, and the definitions of malnutrition and protein energy wasting are still controversial. Furthermore, most nutritional markers and interventions were assessed in hemodialysis patients, while hemodiafiltration and peritoneal dialysis have been less extensively studied. The significant loss of albumin in these two dialysis modalities makes it extremely difficult to interpret common markers and scores. Despite these problems, hemodialysis sessions represent a valuable opportunity to monitor nutritional status and prescribe nutritional interventions, and several approaches have been tried. In this concept paper, we review the current evidence on intradialytic nutrition and propose an algorithm for adapting nutritional interventions to individual patients.

## 1. Introduction

Towards an unrestricted, personalized approach to diet in dialysis patients.

Dialysis and nutrition are two sides of the same coin—dialysis depurates the metabolic waste that is typically produced by food intake; hence, dietetic restrictions are commonly imposed on dialysis patients in order to limit, in particular, potassium and phosphate and avoid fluid overload [1,2,3,4,5,6,7,8,9].

This restrictive approach is being increasingly challenged, however, since it makes the difficult life of dialysis patients even harder, and the generic nutritional restrictions prescribed have significant limitations and side effects (Table 1) [1,2,10,11,12].

The most common foods that nephrologists, nurses, and patients believe should be limited or avoided are those rich in potassium, phosphate, and sodium [3,4,5,7,13,14,15]. However, these restrictions are virtually incompatible with a healthy diet (rich in plant-based food), high protein intake (coupled with phosphate intake), and high calorie intake (associated with varied and good-tasting food) [4,5,6,7,8,9,10].

Furthermore, recent studies have shown that the current food tables do not take additives into account, and that phosphate and potassium content of processed foods may be significantly higher. Moreover, added phosphate or potassium are rapidly bioavailable and potentially more toxic [16,17,18,19,20].

Incremental dialysis, aimed at respecting residual kidney function, and intensive dialysis, aimed at improving efficiency, may be ways to counteract the need for restricted diets, keeping in mind that standard indications apply only to thrice-weekly “conventional” hemodialysis [21,22,23,24,25,26].

However, the risk of depletion of important nutrients during dialysis sessions with highly permeable membranes cannot be ruled out, and, while the advantages of better depuration are clear in young, well-nourished patients, it is still far from certain that high flow hemodiafiltration produces advantages for malnourished patients [26,27,28,29,30,31,32,33].

The definitions of malnutrition and protein-energy wasting (PEW) in dialysis patients is neither simple nor universally agreed upon. Each marker or evaluation proposed has limitations and, as a result, not all clinicians and authors adopt the same markers to assess nutritional status [34,35,36,37,38,39,40,41,42,43,44,45,46,47,48]. 

Even the terms indicating nutritional status in dialysis patients have changed: cachexia, malnutrition, protein energy wasting, and malnutrition inflammation syndrome (MIA) are all used to describe dialysis patients, sometimes with similar meaning, sometimes with different nuances (Table 2) [34,35,36,37,38,39,40,41,42,43,44,45,46,47,48].

Moreover, dialysis prescription varies widely, and very different treatments may be gathered under the same name; information on the dialysis schedule is often incomplete, thus making comparisons and implementing strategies can be challenging [29,30,31].

Long-term interference with lifestyle is a problem. The case of diet and dialysis is a good example of what is involved. Simplistic approaches often lead to failure; delivering more dialysis, giving more food, or adding nutrients in the dialysis session may help some patients, but, once more, one size does not fit all [49,50,51,52,53]. 

Time spent on dialysis represents about 12 h per week, leaving 156 h “unattended”. This time may, however, be precious for controlling, teaching, and, at least to some extent, performing nutritional interventions. Which interventions are more prone to being successful and to what extent is not clear, since clinical situations, comparators, and type of patients are at least as diverse as the interventions themselves.

In this concept paper, following a strategy of high adaptation of dialysis schedules, we present the policy progressively developed in our center and a review of the literature on which it is based. While formal validation is beyond the scope of this paper, we refer, whenever possible, to the initial clinical experiences supporting the choices we describe.

## 2. Intradialytic Nutrition: A Bit of History

When hemodialysis was delivered in 12-h or 8-h sessions using cuprophane membranes, at a time when hyperkalemia was a frequent cause of death, the dialysis session often represented an opportunity for patients to eat more liberally. In a population that was mainly young and had low cardiovascular mortality, except that linked with the effect of long-standing, incompletely corrected uremia, the start of the dialysis session was often the only time a patient could eat an unrestricted meal [54,55].

This was routinely done in home hemodialysis, but many pioneering centers, practicing long-hour dialysis, such as the historic one in Tassin, used to give patients a large, rich meal, usually at the start of dialysis, with the idea that the potentially harmful substances (potassium, phosphate, and salt) would eventually be depurated during the dialysis session [56].

This policy is still customary in the evening shifts in many centers and for patients on long dialysis, in which the depletion of phosphate and water-soluble vitamins may warrant the systematic prescription of nutritional supplements. 

When nephroangiosclerosis and diabetes—rather than pyelonephritis and glomerulonephritis— came to be seen as the causes of end-stage renal disease (ESRD), this was not without consequences for dialysis tolerance, and the first caveats on eating during dialysis date to this phase. The progressive shortening of dialysis time, made possible by increased dialysis efficiency, occurred at the same time as the dialysis population started to become progressively older, a process that is continuing, at least in many European settings [57,58,59,60,61,62].

Furthermore, in the eighties, the decision to shorten dialysis time to 4 h or less led to the widely-held idea that there was no point feeding patients during dialysis since, in any case, their nutritional status depended solely on their overall dietary habits. For a number of years, economic and logistic constraints led to intradialytic interventions being seen as a sort of “hotel” service, more aimed at interrupting the long dialysis session than addressed to a specific nutritional advantage. However, some centers continued to allow patients to eat banned foods, such as those rich in phosphate (cheese and ham) or those rich in potassium (bananas and other fruit), partly as a way to partially compensate for patients’ usual dietary restrictions.

Given growing concerns about malnutrition, the emerging importance of non-compliance with dietary prescriptions and the logical albeit simplistic conviction that malnutrition in dialysis patients could be countered by increasing protein and calorie intake, the dialysis session once again came to be seen as a favorable moment for effective nutritional interventions. Although some authors still claim that dialysis patients should not eat during treatment, this policy is increasingly being abandoned, and oral supplementation and intravenous intradialytic nutrition are increasingly being practiced with a variety of outcomes in different populations [63,64,65,66,67].

While intravenous nutrition was once usually limited to cases of severe malnutrition, the use of oral intradialytic nutrition as a means of preserving long-term nutritional status is being reported in a growing number of studies.

## 3. From the “Skeleton Man” to the Obese Sarcopenic Patient

In the collective imagination, when we think about malnutrition on dialysis, the “skeleton man” is probably the first image that comes to mind; this evocative and somehow crude definition dated to the early dialysis era in which cachexia was frequently an effect of under-dialysis in a young person with high metabolic needs. The combination of insufficient depuration, by itself exerting a catabolic effect, and of multiple alimentary restrictions led to protein energy wasting, resulting in a metabolic disaster [68,69,70].

At least in settings where dialysis efficiency has been a key goal in recent decades, the “skeleton man” is now seldom encountered in dialysis wards, except in cases in which diffuse and severe vascular disease is accompanied by a reduction in food intake, often in a context of cognitive impairment in elderly patients [71,72,73,74]. Another exception is patients with a very long dialysis follow-up, whose clinical situation is often dominated by diffuse precocious bone or vascular disease [75,76,77,78,79,80] (Figure 1). 

However, it is not unusual to find that these thin, fragile patients have very high dialysis efficiency, often linked to their low BMI and volume distribution, whose dialysis prescription should probably be decreased rather than increased [29,30]. These cases, particularly the Japanese school, suggest modulating dialysis efficiency to avoid loss of precious nutrients.

At present, the type of malnutrition most commonly encountered, at least in European settings, is sarcopenic obesity [81,82,83]. Although overweight or obese, patients present a severely reduced muscle mass, often with preserved calorie and protein intake (Figure 2). These cases are difficult to label; they are not malnourished in the usual sense, they do not have energy wasting, their BMI is normal, and their calorie intake is usually normal as well. Often, they do not have clear signs of malnutrition/PEW according to the standard parameters, such as prealbumin or transferrin levels, and their albumin levels may be modulated more by type of dialysis than by other means. Yet, they have a severely reduced muscle mass, and this reduction is correlated with survival [81,84,85,86,87].

This observation, as is discussed later in this paper, is leading some experts to supplement nutritional therapy with interventions aimed at preserving muscle mass and combining intradialytic nutrition and intradialyitic exercise, seen to be the (missing) factor capable of transforming a relatively static model (nutrition vs. dialysis) into a dynamic one [88,89,90,91,92,93].

## 4. Intradialytic Nutrition: Pros and Cons

As mentioned, it is never easy to deal with the dialysis population. In addition to the potential side effects associated with intradialytic nutrition, its use is ill-advised in some situations, and there is no consensus on the best way to deliver it (in cases where it appears to be a viable alternative). A severe lack of reliable information makes the prescription of intradialytic nutrition even more empiric than prescription of the dialysis session itself. 

## 5. The Context and The Concept: A Tailored Approach to Dialysis Prescription

The concept on which this paper is based can be summarized as a tailored approach to intradialytic nutrition based upon the patient’s clinical characteristics and nutritional status. This “ladder approach”, depicted in Figure 3 and discussed below, is the logical complement of a tailored approach to dialysis prescription based on life expectancy, vascular access function, and nutritional status [30]. Interestingly, a paper on a similar approach was recently published by Japanese nephrologists known to pursue a “softer” hemodiafiltration schedule than the traditional French one [29].

In both cases, well-nourished patients are dialyzed with high-performance schemes, whose primary goal is reaching high efficiency. Conversely, dialysis intensity is reduced, and treatment schedules are “tempered” in very fragile patients for whom tolerance becomes the main goal (Figure 3).

## 6. Concept and Context: A Special Case, Incremental Hemodialysis

The idea that, in dialysis, “more is better” is not true for fragile patients. The failure of early dialysis start, which was previously considered synonymous with “healthy” dialysis start, to improve patients’ prognoses, the recognition of the importance of residual kidney function in determining survival, and the association of the latter with better nutritional status are all elements in favor of a progressive, incremental dialysis start aimed at reducing dialysis-related morbidity and mitigating “dialysis shock” [21,22,23,94,95,96,97,98,99].

There is disagreement about the definition of incremental dialysis regarding the number of sessions there should be per week and what their scheduling and sequence should be (starting with shorter sessions, increasing frequency first and duration later or starting with fewer sessions of longer duration). Consequently, nutritional indications must once more be contextualized and adapted [21,22,23,99,100,101,102]. 

As for nutrition, two almost opposite options are reported in the literature. The first holds that, regardless of dialysis frequency, all patients who start dialysis should be counseled to follow a high-protein diet to combat the ever-present threat of malnutrition; the second progressively liberalizes the diet as the number of dialysis sessions increases [103,104]. Each approach has a logical basis, but some evidence supporting the second option comes from Italy, where incremental dialysis was recently reintroduced, and nutritional predialysis care is widespread [16,103,105]. Patients often start dialysis after following a moderately protein-restricted diet, a policy that is not usual in other European countries and the United States. In these countries, incremental dialysis prescription is aimed principally at favoring adaptation by the patient to the new treatment, both from clinical and psychological points of view. While, in the first case, progressive liberalization of the diet may be seen as a partial compensation of prior protein restriction, in the second one, reducing protein intake would only add to the patient’s difficulties in complying with the new, already intrusive treatment. 

## 7. The “Ladder Approach” to Intradialytic Nutrition

Our basic idea was to adapt a stepwise approach in which nutritional supplements or intradialytic food would be managed with a larger choice for well-nourished patients, up to intravenous supplementation of albumin, in selected cases with severe hypoalbuminemia and a catabolic status. Nutritional improvement would make it possible to go back from intravenous to oral supplements and regular food (Figure 3). The better the patient’s nutritional status or the more improvement being shown, the more the patient is allowed to determine their dietary choices (Figure 3).

## 8. Eating “Normal” Food During Dialysis

Natural solutions are usually better than artificial ones; this seems also to be the case for the choice of modulating the intake of regular food for hemodialysis patients, taking advantage of dialysis sessions to combine nutritional interventions and physical rehabilitation. 

The decision to use regular food has two main advantages: allowing patients to choose what they actually like best (thus making the usually far from enjoyable dialysis routine a bit more bearable) and lower cost. Furthermore, choosing food wisely is more in line with a healthy dietary approach, taking into account not only the quantity of nutrients that are consumed but also their quality, and avoiding preserved and modified foods.

The experiments reported in the literature are protean, and the foods included vary. They seem to have been chosen mainly for their high protein content (for example, a preparation based on egg white in a recent Chinese study) or, less frequently, because of their cultural acceptance and higher palatability, as in a Brazilian experiment in which patients were given red meat snacks during dialysis [65,66,67,68,106,107,108,109,110]. Combination with phosphate binders is sometimes a choice, similarly to what is advised in phosphate-rich meals for people who are not on dialysis [108].

Whatever the reasons for choice and the food offered, eating during dialysis sessions seems to do more good than harm, provided the patient is not hemodynamically instable and that dry weight is carefully assessed to avoid hypotension, nausea, and vomiting [64].

According to our algorithm, modulation of intradialytic meals or snacks is the first step to preserving nutrition in patients with a good nutritional status and to improving it in those cases in which initial signs of protein energy wasting or sarcopenia are likely to be reversible with combined nutritional and dialysis approaches integrated wherever possible by physical exercise (Figure 3).

Following this hypothesis, in our center, we attempted to improve compliance by offering patients a greater variety of food choices and adopted a combined nutritional and educational approach aimed at improving protein and calorie intake as far as possible in patients with normal or moderately increased BMI as well as the protein intake in those with obese sarcopenia. Patients are allowed to choose intradialytic food from a varied “menu” of snacks that are rich in protein and/or calories (Figure A1 in the Appendix A).

There are many cultural, economic, and logistic barriers to what is the simplest and probably the safest way to increase food intake and combat malnutrition. Culturally, in a medicalized society, the use of “normal food” may not be seen as medical treatment, and it may be difficult to convince the hospital management to offer dialysis patients a rich choice of snacks and meals. Furthermore, the logistics of a dialysis ward are not always suitable for preparing snacks for patients, and food management may not be one of the tasks included in the routine activities of a dialysis center. The decision to let patients bring their own snacks is feasible, but this policy may not work for those patients who are most in need (elderly, living alone, anorectic, etc.). Economic issues are often raised, yet food is usually less expensive than oral nutritional supplements, and our opinion is that the organizational problems can be overcome with little economic burden and great advantage for patients’ well-being. 

## 9. Oral Nutritional Supplements

The idea that fragile or elderly patients need assistance when they eat and that highly concentrated food supplements can facilitate intake of needed nutrients appears to be a reasonable one; these supplements are rich in proteins and calories, well standardized, and do not require preparation. Furthermore, a rich formulation is usually concentrated in a small amount of liquid or is semisolid. This is an advantage for patients who are anorectic, unable to eat a large amount of food, or have no specific food preferences, so that giving them oral supplements is an easy way to ensure that they obtain the recommended quantities of calories and proteins (Table 3). 

Many dialysis units now offer these oral supplements, at least to hypoalbuminemic patients, and a number of studies have addressed their effect on nutritional status and albumin or prealbumin levels [111,112,113,114,115,116,117]. Overall, reports are favorable on all outcomes tested; an interesting suggestion is the fact that providing oral nutritional supplements during dialysis seems to reduce the number of missed dialysis sessions, thus underlining the psychological importance that nutritional care has for our patients [116].

However, these advantages are at least partly counterbalanced by the presence of side effects. Most of the available nutritional supplements are sweet, and this may be a problem for those patients who do not like sweet-tasting food and who, in the long run, tend to avoid them. The high energy content may have the effect of limiting the intake of normal food. Furthermore, in most commonly available formulations, phosphate is not restricted; there are no restrictions on additives and taste enhancers whose toxicity is only partially known and could be higher in dialysis patients. No less importantly, they are expensive (in France between €2 and €5 per supplement), and the price is usually higher for the more specific formulations needed by dialysis patients.

In our center, during dialysis, we include a choice of oral supplements along with snacks (Figure A1 in the Appendix A). We try to orient patients with lower albumin levels towards protein-rich options while remembering that personal choice and dietary variation are fundamental for attaining compliance. 

From a nutritional point of view, dialysis induces a negative nitrogen balance, hence oral intradialytic supplements are also an option for well-nourished patients, compensating for the loss of proteins and amino acids during hemodialysis sessions. In addition to increasing protein intake, other options for reducing amino acid losses during dialysis include higher carbohydrate intake, inducing insulin secretion, and lowering amino acid circulating levels and hence their loss in dialysate [118,119,120].

## 10. The “Case” of Ketoacid and Amino Acid Supplementation

Amino acid mixtures have been employed in a number of studies to enrich the protein content of the food dialysis patients are offered; their advantage, if any, over other options is not clear, and logistics is probably the most important factor in preferring this approach [121]. 

In the pre-dialysis phase, plant based, low, or very low protein diets supplemented with a mixture of essential amino acids and ketoacids are generally associated with a slower progression of chronic kidney disease (CKD), and some data on the chronic use of these supplements in dialysis patients are promising with respect to the preservation of residual kidney function and urine volume output [122,123,124,125]. We therefore consider the use of this formulation in dialysis patients with low protein intake and low albumin levels and who prefer “pills” to other oral supplements.

The only formulation presently available in France is Ketosteril, which consists of film-coated tablets containing ketoanalogues in forms of calcium salts—a-ketoanalogue of isoleucine, leucine, phenylalanine, valine, methionine—and the following amino-acids: lysine, threonine, tryptophan, histidine, and tyrosine. The usual dose, as a complement of low protein diets, is one tablet each 5–10 Kg, according to the level of protein restriction. The dose in dialysis patients is not clearly stated; the addition to intradialytic snacks is now under evaluation. While the drawback is the addition of “more pills” to our patients’ already large tablet intake, the advantage is they are less likely to interfere with appetite, thus possibly better preserving spontaneous food intake. 

## 11. Intravenous Formulations

The dialysis session is a good opportunity for intravenous administration, useful from the nutritional point of view when food intake is inadequate. A number of studies, in fact, suggest that, in selected patients, intradialytic intravenous nutrition can improve the nutritional status of malnourished dialysis patients in the short-medium term [126,127,128,129,130]. 

Once more, things are not as simple as they seem; adding intravenous nutritional fuel to a malfunctioning metabolic machine is not always effective (Table 4). On the one hand, the utilization of proteins may be impaired by a shut-off anabolic phase, and amino acids may reduce appetite and increase urea and acidosis if not correctly utilized. Sugars may aggravate the glucose intolerance encountered in dialysis, and the role of triglycerides, often added as calorie fuel to the intravenous mixture, in worsening atherosclerotic lesions is not clear. Nausea, muscle pain, infections, hyperglycemia, and complications linked to the dialysis procedure have occasionally been reported [126,127,128,129,130,131,132,133,134,135,136]. While baseline dyslipidemia is a contraindication for treatment, no target level based on an evidence-based position has yet been fixed [128].

There are several available intravenous preparations differing in concentration and nutrient balance. On average, an intradialytic infusion of about 1 L of intravenous supplement provides 800–1200 kcal in the form of glucose, lipids, and amino acids, which, on average, are between 30 and 60 g per liter; however, the quantity of nutrients lost in the dialysis procedure is not at all clear [128]. While administration is usually limited to the last hour(s) of dialysis, when the dialysis membrane is probably less permeable, due to the “protein cake” that partly saturates its capacity to dialyze molecules of higher molecular weight, we were not able to retrieve any comprehensive study on this question. Their cost, estimated as 10 to 100 times higher than oral supplements, further limits their use, at least in settings where these supplements are not reimbursed [129]. 

We therefore suggest that that these supplements can be useful in the short-medium term for patients that, in spite of low food intake, may be able to reverse the metabolic balance from catabolism to anabolism. Naturally, their limits need to be acknowledged, as does our lack of sound evidence in favor of any change in hard outcomes [126,127,128,129]. Furthermore, since the losses are higher in hemodiafiltration, when specific techniques including endogenous reinfusion are not available, the advantages of better depuration of the middle molecules including those involved in inflammation have to be weighed against the use of hemodialysis so that the metabolic advantages, particularly those of amino acid supplementation, are not lost [130,131].

## 12. Albumin (And Blood Transfusions)

When a dialysis patient is under profound metabolic stress, as in the case of an acute disease or surgery, a rapid, drastic reduction in serum albumin is often observed, together with a dramatic reduction in prealbumin and hemoglobin. These alterations, often accompanied by profound deterioration of general well-being, are not likely to respond to an increase in dietary intake or to an adaptation of the dialysis schedule. The low prealbumin level is evidence of low metabolic reactivity and is usually closely linked to the patient’s inflammatory status and overall clinical condition. Not surprisingly, in these acute phases, in which the metabolic machine has shut down, intradialytic infusion of amino acids and lipids has little effect on nutritional status [126,127,128,129]. 

Similarly, blood transfusions are occasionally needed when, in spite of an increase in erythropoiesis stimulating agents and the correction of vitamin deficits and iron stores, a patient remains severely anemic. This is most frequently observed in the same patients that fail to respond to dietary or intravenous supplementations. In these cases, most commonly in the context of acute or chronic inflammatory states and the malnutrition-inflammation-atherosclerosis syndrome, transferrin saturation and reticulocyte count are low in the presence of high ferritin levels and are evidence of a lack of metabolic response [132,133,134,135].

The pattern of low albumin level with a severely depressed prealbumin level is frequently concomitant. The meaning of infusion of albumin during dialysis in this context is similar to the infusion of albumin in the case of ascites drainage in cirrhotic hypoalbuminemic patients. Although the practice of albumin infusion in severely depleted dialysis patients is probably more frequent, there are very few studies that address this question. [136,137]. As in many cases of severely ill patients, both beneficial and negative effects are difficult to ascertain. Once more, the problem is the same as the one encountered when using albumin for cirrhotic patients, a practical choice that is not fully supported by clear-cut evidence.

While there are even fewer evidence-based studies on albumin infusion in dialysis patients, the few available ones suggest, as is logical to expect, that albumin infusion may have a favorable effect on intradialytic hypotension and a low risk of side effects. However, the advantages in the management of hypotension compared to saline infusion are not clear and probably cannot be generalized [135,136,137,138,139,140]. The high costs, the potential side effects, and, in particular, the risk of allergic reactions, need to be underlined. 

In our experience with a high comorbidity population, we have employed albumin infusion in highly selected cases with low albumin, low prealbumin, and high inflammatory markers, usually following surgery or severe infection. In most of these cases, hypotension was symptomatic and impaired the attainment of correct depuration. In this context, no allergic reactions were observed during over 1000 dialysis sessions in which infusion of albumin was practiced over three years of activity (about 50,000 dialysis sessions), possibly also because of the anergic state that usually accompanies severe malnutrition.

Albumin infusion is now being increasingly performed in the context of rheopheresis coupled with dialysis, an emerging technique for the treatment of severe vascular diseases including calciphylaxis, a disease typical of fragile dialysis patients with a combination of risk factors including immunologic diseases, inflammation, sarcopenic obesity, atherosclerosis, underdialysis, and deranged calcium-phosphate balance [141,142]. 

## 13. Are We Doing All We Can? The Role of Intradialytic Exercise

The missing element in the definition of the metabolic pattern of dialysis patients is physical activity. Metabolic balance is, in fact, significantly affected by physical exercise, which is a well-known modulator of the anabolic versus catabolic balance in kidney disease patients [143,144,145,146,147,148].

To be effective, nutrient intake must administered in an anabolic setting, and physical exercise is the only modifiable anabolic stimulus for muscle tissue. 

While an in-depth discussion of the potential role of intradialytic exercise in this context is beyond the scope of the present review, the association of physical activity should probably be proposed on a similar scale, from rehabilitation exercises for fitter individuals to mild exercise during dialysis for those who are more fragile and have less support at home and to passive exercise for the patients who are in poor clinical condition [149,150,151,152,153,154,155,156]. 

Clinical limitations, economic constraints, and cultural nihilism have been significant barriers to the systematic implementation of physical exercise in the dialysis ward, but the concept that physical activity is a necessary complement of nutritional management is increasingly being accepted and should be borne in mind when setting up a nutritional support program in a dialysis ward [46].

## 14. What This Review Did Not Address

This review did not try to define malnutrition or the best way to monitor it, limiting discussion to controversial practical clinical points [157,158,159]. The issue of diagnosing and staging malnutrition would need a dedicated review paper of its own. To give an idea of its complexity, Table 5 summarizes some of the different means commonly used to evaluate nutritional status in dialysis patients.

Furthermore, the review does not address enteral nutrition, since this choice is usually proposed not only during the dialysis session and, in our experience, the indication for enteral nutrition came from problems not strictly related to the dialysis performance. While this approach seems to be associated with some improvement in different studies, we believe that, at least in our setting, it is very poorly accepted by patients, and this low cultural acceptance might negatively interfere with other nutritional strategies [160].

Another question the review did not address is the use of anabolic drugs, such as steroids or growth hormones, limiting the discussion to “non-pharmacologic” nutritional interventions. The use of anabolic steroids can indeed be considered in selected cases, at least for short periods, but the potential side effects should limit the prescription of these non-routine interventions to particular cases [50,161,162,163,164].

Another suggestion is the use of β-hydroxy-β-methylbutyrate (HMB), a metabolite of leucine able to counteract protein degradation, increase protein synthesis by inhibition of proteasome and stimulation of mTOR pathway, increase the number of mitochondria, and reduce pro-oxidant and inflammatory status. Improvement of sarcopenia and preservation of muscle function are potential effects of HMB supplementation [165,166,167]. While studies performed in fragile or sarcopenic elderly people are promising on muscle mass and on muscle strength and function, studies in dialysis patients are lacking. A double blind, placebo-controlled, randomized trial of daily HMB supplementation up to six months induced no significant advantage in dialysis patients, but a larger scale trial is needed [167]. Intradialytic creatine supplementation has also been proposed as an alternative to improve muscle mass and well-being. Once more, there is need for large studies to validate these potentially revolutionary approaches [168,169].

While acknowledging that malnutrition is usually linked to a patient’s comorbidity and not to the simple lack of alimentation and underlining the importance of inflammation in the development of clinical malnutrition in our patients, this review does not touch on the issue of how to avoid or reverse the dialysis-related inflammatory syndrome, which is clearly one of the main, often unmet goals of treatment for chronic dialysis patients [170].

## 15. Final Considerations

Restoring or preserving good nutritional status in dialysis patients is not easy, since what can broadly be called malnutrition is usually the result of complex metabolic derangements as well as of severe comorbidities. 

There are several non-nutritional reasons for this particular dialysis-related syndrome of which malnutrition or protein energy wasting are only one aspect, while inflammation, comorbidity, lack of physical exercise, and insufficient dialysis all contribute to determining the clinical picture. It is therefore highly unlikely that a simple increase in food intake will be a panacea. 

While nothing is simple in dialysis, the fact that patients are seen three times per week during treatment sessions leaves room for tailored interventions. 

The present concept paper suggests employing a stepwise approach against malnutrition in dialysis, starting from nutritional counseling, trying to increase the intake of healthy food, exploiting the potential of intradialytic snacks, followed by nutritional supplements, choosing between different combination, favoring snacks rich in energy or in proteins, upgrading to intradialytic intravenous nutrition, and leaving albumin infusion as a last resort in patients who are metabolically non-reactive and severely malnourished patients. Ideally, albumin and intravenous nutrition should be discontinued after nutritional improvement, with the final aim of downgrading the intervention to oral supplements and eventually to increased quantity of healthy food. Employing all the available means in a patient-adapted and, whenever possible, a patient-friendly way may help to slow the downwards nutritional and clinical spiral often observed in dialysis patients.

## Figures and Tables

**Figure 1 nutrients-12-00785-f001:**
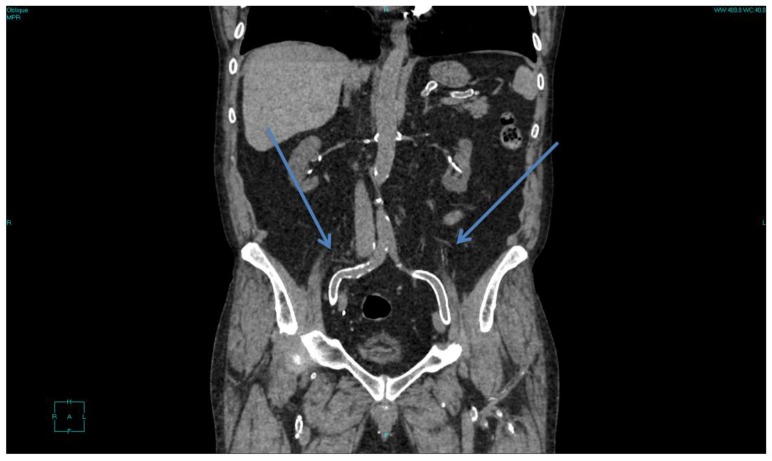
Diffuse vascular calcifications in a patient with more than 30 years of follow-up on dialysis and after kidney transplantation. Arrows show eggshell calcifications of the iliac axes; scattered calcifications are visible in all other districts.

**Figure 2 nutrients-12-00785-f002:**
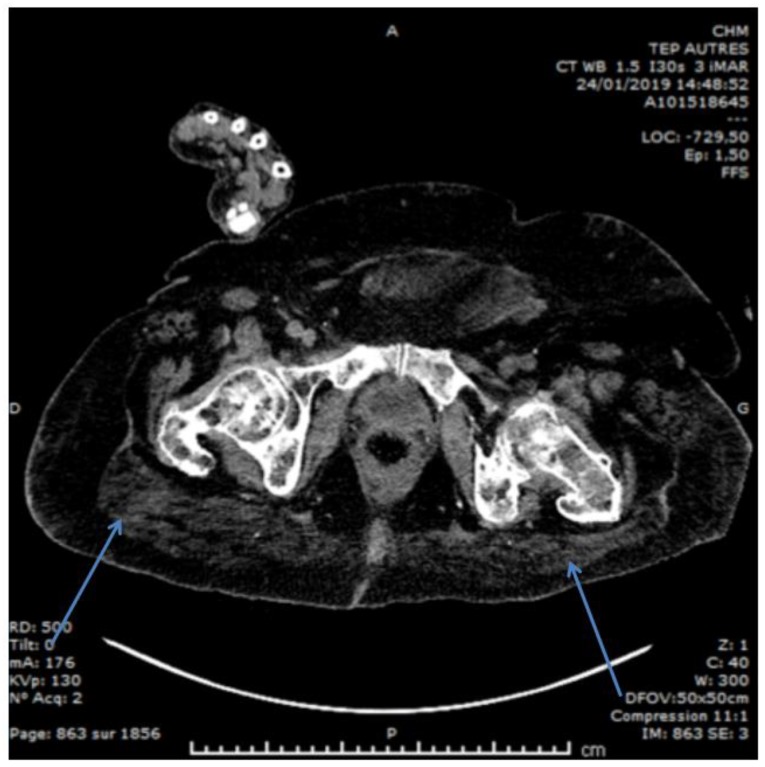
An example of sarcopenic obesity in a chronic hemodialysis patient affected by multiple myeloma. Arrows show poor quality, muscle tissue enrobed by well-developed adiposity.

**Figure 3 nutrients-12-00785-f003:**
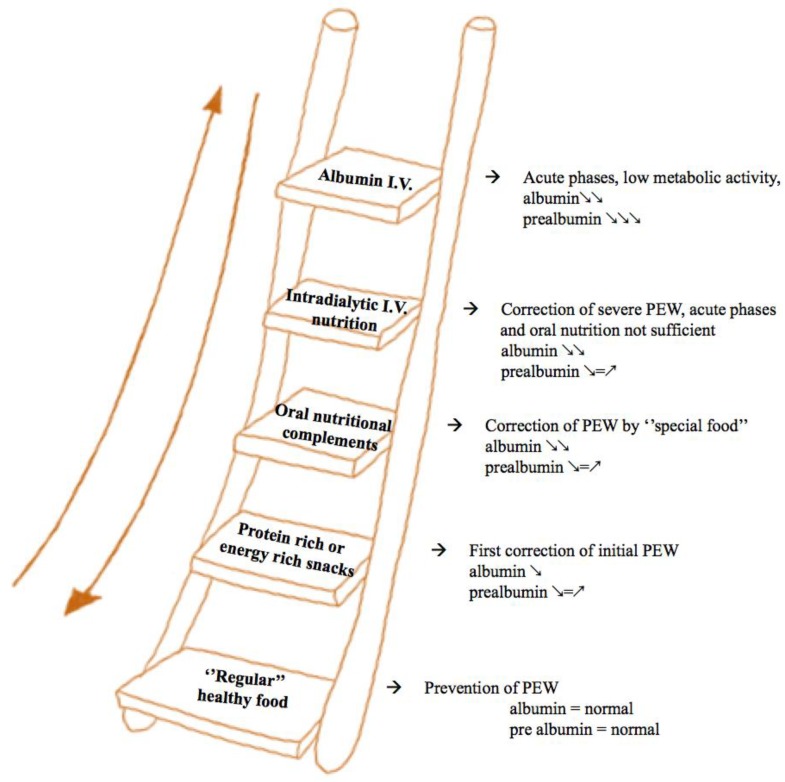
A ladder, stepwise approach to interdialytic nutrition. In the first steps of the ladder, when nutritional markers are still normal, healthy food is advised to maintain nutritional status. Oral complements are needed when appetite is low and when energy and protein dense food may help restoring the nutritional balance. Intradialytic i.v. nutrition is an option in phases in which oral alimentation is low but the metabolic machine is not shut off, as witnessed by normal pre-albumin levels. Albumin may be needed when the low prealbumin levels suggest a very limited anabolic potential, not simply due to the lack of substrates, but usually linked to inflammation of acute diseases.

**Table 1 nutrients-12-00785-t001:** Limiting food versus allowing liberal nutrition in dialysis patients.

	Rationale	Pros	Cons	Controversial and Unclear Issues
Salt (and water)	Limits weight gain, improves hypertension control and dialysis tolerance.	Low sodium intake is feasible and associated with better blood pressure control and dialysis management.	Limiting salt may reduce palatability and induce anorexia; it may not be an option for patients living in retirement homes, or receiving cooked meals at home.	How to manage Na and water restriction (if any) in patients with residual kidney function.
Potassium	Limits the risks of hyperkalemia.	Potassium is derived from diet, and its reduction in the diet can reduce the risk of hyperkalemia.	K restriction is commonly interpreted as reduced consumption of fruit and vegetables, which are associated with better cardiovascular outcomes.	The missing factor is potassium absorption, which may be enhanced in the case of slow intestinal transit, enhanced by a diet poor in fibers and use of potassium binders.
Phosphate	Counterbalances CKD-related metabolic bone disease.	Phosphate levels are associated with vascular calcifications; a high phosphate level is cardiotoxic and is a stimulus for PTH secretion.	Phosphate content is higher in protein-rich food; therefore, too strict a reduction can be incompatible with high protein content.	The issue of phosphate added to food is only partially known. The role of additives may be more important than previously appreciated.
Lipids	Counterbalances cardiovascular risk and accelerated atherosclerosis in dialysis patients.	Dyslipidemia is a common finding in dialysis patients; nutritional interventions should always come first.	Lipids are important sources of energy. Restriction should be balanced against the indication for high energy intake.	The role for statins in dialysis patients is controversial; physical activity may be an important non pharmacologic aid to control dyslipidemia.
Carbo-hydrates	Counterbalances carbohydrate intolerance of uremic patients.	In several dialysis settings, more than half of the patients are diabetic; carbohydrate intolerance is commonly associated with worse outcomes.	Carbohydrates are important sources of energy. Restriction should be balanced against the indication for high energy intake in dialysis patients.	Physical activity may be an important non pharmacologic aid to improve the overall metabolic balance.

Legend: CKD: chronic kidney disease; PTH parathyroid hormone.

**Table 2 nutrients-12-00785-t002:** Some definitions of “malnutrition” in dialysis patients.

Definitions	Advantages	Limits
Malnutrition	Intuitive and comprehensive; replaces the obsolete term “denutrition”, highlighting the importance not only of quantity of food but also of its distribution and quality.	The definition has changed over time, and the term is usually employed to describe a combination of muscle wasting, low nutrient intake, and low nutritional markers. Its meaning in the context of “poor quality nutrition” is often lost. Generic, too often based on albumin levels, now recognized as only one of the markers of malnutrition in dialysis patients (interference with mode of dialysis and inflammation). Focuses attention on intake, and less on causes of wasting (see below)
Wasting	Proposed in 1983 by the World Health Organization to define an involuntary loss of weight of more than 10% in absence of specific diseases such as opportunistic infection, cancer, or chronic diarrhea.	Generic, the time of development is not univocally defined. Probably more able to describe rapid changes, which are not the most frequent in dialysis patients, in which the process is often long. Focuses attention on intake, which may be the result and not the cause of an underlying process.
PEW: Protein-Energy Wasting	Widely used. proposed by the International Society of Renal Nutrition and Metabolism. Focuses attention on the relationship between malnutrition and metabolic background.	Often only considers albumin and cholesterol; may be biased in hyperlipidemia patients and does not account for the causes of low albumin levels. Includes BMI, but obese sarcopenia may be overlooked.
MIA: Malnutrition Inflammation Atherosclerosis syndrome	Focuses attention on the relationship between malnutrition, inflammation, and comorbidity. It is a dynamic index and is sensitive to variations in clinical status.	Relies, among other indexes, on subjective evaluations, which may be operator-dependent. Includes BMI, but obese sarcopenia may be overlooked.
Cachexia	According to the Society on Sarcopenia, Cachexia and Wasting Disorders (2008), cachexia is a complex metabolic syndrome associated with underlying illness and characterized by loss of muscle, with or without loss of fat. According to the International Society of Renal Nutrition and Metabolism (2008), cachexia is the last stage of PEW.	The different definitions make systematic use of this term difficult. The Society on Sarcopenia, Cachexia and Wasting Disorders has introduced the concept of fatigue, which may be misleading in severe chronic diseases and in elderly patients.

**Table 3 nutrients-12-00785-t003:** Open issues in oral nutritional supplementation in dialysis.

	Pros	Cons- Unclear
Interference with depuration	Concomitant dialysis may make it possible to reduce the risk of fluid overload; if meals or snacks are given at the start of treatment, excess phosphate or potassium can be removed during the dialysis session.	We lack data on interference with dialysis efficiency. Low tolerance (hypotension) can lead to shortened dialysis time, or reduced blood flow and dialysis efficiency.
Long-term effects	Small studies report good results in selected patients.	Long-term advantages are not clear in pooled data, possibly due to the heterogeneity of indications and populations.
Tolerance	Good, unless the patient develops hypotension during or immediately after the meal.	Old studies suggest withholding food during dialysis. However, high- protein, high-fat meals were often supplied, and weight loss was often considerable.
Losses during dialysis	Probably minimal.	No clear contraindication.
Additives and preservation agents	Widely used in industrial food processing to reduce contaminations and enhance duration.	Very little studied; while phosphate and potassium containing additives are usually avoided less is known about other substances and trace elements. This is a question that needs further study.

**Table 4 nutrients-12-00785-t004:** Some open issues in intradialytic parenteral nutrition.

	Pros	Cons- Unclear
Interference with depuration	Concomitant dialysis makes it possible to reduce the risk of fluid overload.	We lack data on interference with dialysis efficiency. Low tolerance may lead to shortened dialysis time.
Tolerance - contraindications	Tolerance can be regulated by management.	Tolerance may be low (hyperosmolar media). Dyslipidemia is a reported contraindication, but few studies select for this item.
Losses during dialysis	The metabolic balance is positive in clinical studies.	The quantity lost is not clear; the use of parenteral nutrition mainly in the last hour(s) of dialysis can reduce loss, but interaction with the dialysis membranes is not clear.
Prescription modalities	Different products are available, potentially allowing personalization of treatment.	Experience with “nonconventional dialysis” is minimal. Re-feeding can be a life-threatening problem in dialysis patients.
Short-term effects	Small trials report good results in selected patients. These have not been confirmed in large meta-analyses.	The metabolic machinery needs to be at least partially preserved to make it possible to exploit the anabolic potential of the substrates. This may not be the case for patients with acute problems or severe inflammation, in which excess non-metabolized proteins may increase acidosis and urea levels.
Long-term effects	Small trials report good results in selected patients. These have not been confirmed in large meta-analyses.	Long-term effects are not clear in pooled data, possibly due to the heterogeneity of indications and populations. Some studies report an even higher mortality rate in patients treated with intradialytic nutrition.

**Table 5 nutrients-12-00785-t005:** Nutritional evaluation in dialysis patients: some advantages and limits of the common tests.

	Suggested Frequency	Advantages	Limits
Anthropometry			
Body weight (b.w.) Middle arm circumference Triceps skinfold thickness Skeletal muscle circumference	Each treatment Monthly Monthly Monthly Monthly	Non-invasive, provides immediate results, easily compared in different settings.	Precise measurements need a skilled operator and are relatively demanding in terms of time.
Body composition			
Bio-impedance analysis (BIA)	Monthly for body composition up to each treatment to evaluate fluid overload.	Non-invasive, provides immediate results, easily compared in different settings.	BIA should be performed at least 15 min after the end of the dialysis; patients may be reluctant to wait; the cost of the electrodes is relatively high. Difficult to standardize in patients with amputation or skin problems. Has to be interpreted with caution in obese of anorectic patients.
Biochemical data			
Serum albumin Total proteins Transferrin Prealbumin Glucose Lipids Protein nitrogen Appearance (PNA) Lymphocyte count Liver enzymes	Monthly	Valuable tools to assess effective dietary intake and adherence to dietary prescriptions	All the main nutritional markers are affected not only by the nutritional status but also by dialysis efficiency, type of dialysis (hemodialysis vs. hemodiafiltration) and by the inflammatory status. Interpretation may be difficult, particularly in patients at high comorbidity.
Evaluation scales			
Subjective Global Assessment (SGA) Malnutrition Inflammation Score (MIS)	Quarterly Quarterly	Widely used assessment tools for dialysis patients, useful to compare different series.	SGA is very sensible to rapid changes, may be less sensitive to chronic changes. MIS is a mixed marker, taking into account comorbidity and inflammation, The specific weight of nutrition may be difficult to enucleate.
Dietary habits			
Dietary recall (usually 24 h) Three days or seven days dietary journal Food frequency questionnaires	At least monthly	The evaluation of dietary habits is the first step to evaluate nutritional intervention as gives qualitative and quantitative information to target nutritional counseling	The recall may be biased or difficult in patients with cognitive impairment. Compliance to dietary journals may be difficult. Food frequency questionnaires are often very sensitive to the cultural context and may be difficult to adapt to a multiethnic population.
Functional Tests			
Barthel Index Karnofsky Index	Quarterly Quarterly	Highlight the effect of nutritional status on functional abilities	Indirectly evaluation of nutritional status. Sensitive to the burden of comorbidity.
Performance Tests			
6 min walking test 30′ Sit-to-stand-to-sit Hand-Grip test	Quarterly Quarterly Quarterly	Useful to monitor the effects of a nutritional intervention; hand-grip test is increasingly used to evaluate force as an indirect measure of muscle mass.	The tests are reliable only in experienced hands. Hand grip tests may be performed in different ways, and may be affected by the presence of an arterio-venous fistula or graft.

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
