# Peer review of "Intradialytic Nutrition and Hemodialysis Prescriptions: A Personalized Stepwise Approach"

_nutrients, 2020, doi:10.3390/nu12030785_

Round 1

Reviewer 1 Report

-

Author Response

Thank you for your appreciation.

The authors

Reviewer 2 Report

Summary: The authors present a well researched review of the existing literature related to intradialytic nutrition and summarize their approach to the problem. As the authors state, malnutrition in dialysis patients is a vexing issue, complicated by the lack of standardized definitions and treatments. The authors clearly state the limitations of their manuscript and summarize many of the important questions that have not been answered by the current scientific literature.

Major criticisms:

Abstract: The abstract is overly verbose and as a result, the authors are likely to lose many readers before they get to the substance of the paper. A more succinct and focused abstract would help to increase interest. 

Introduction: While the introduction is very informative and well referenced, the presentation of the information could be enhanced by eliminating unnecessary verbiage. 

The rest of the text presented in a logical manner, although some of the terminology used is idiosyncratic. 

Figures: The manuscript contains five figures.

  Figures 1 and 3 do not add significant information and can be removed. 

  Figure 5 could be presented as an appendix at the end of the manuscript.

Tables

  Tables 1 and 2 are well presented summary. Minor criticism: Salt and water restriction, "limits weight gain, improves hypertension and tolerance". It is unclear what tolerance means. 

   Tables 3 and 4: I do not believe that these tables are essential to the manuscript as most of the information has been included in the text.    

Minor criticisms

Ketosteril may not be familiar to many readers and as such, should be briefly described. 

Author Response

First of all we would like to thank the reviewer for the time dedicated at improving our study.

Secondly, we tried to answer to all the comments and criticisms, as detailed in the following point by point answers.

Abstract: The abstract is overly verbose and as a result, the authors are likely to lose many readers before they get to the substance of the paper. A more succinct and focused abstract would help to increase interest. 

We tried to shortened the abstract as follows:

Dialysis and nutrition are two sides of the same coin: dialysis depurates metabolic waste that is typically produced by food intake. Hence, dietetic restrictions are commonly imposed in order to limit potassium and phosphate and avoid fluid overload. Conversely, malnutrition is a major challenge and, albeit to differing degrees, all nutritional markers are associated with survival. Dialysis-related malnutrition has a multifactorial origin, related to uremic syndrome and comorbidities, but also to dialysis treatment. Both an insufficient dialysis dose and excessive removal are contributing factors. It is thus not surprising that dialysis alone, without proper nutritional management, often fails to be effective in combatting malnutrition. While composite indexes can be used to identify patients with poor prognosis, none is fully satisfactory, and the definitions of malnutrition and protein energy wasting are still controversial. Furthermore, most nutritional markers and interventions were assessed in haemodialysis patients, while haemodiafiltration and peritoneal dialysis have been less extensively studied. The significant loss of albumin in these two dialysis modalities makes it extremely difficult to interpret common markers and scores. Despite these problems, haemodialysis sessions represent a valuable opportunity to monitor nutritional status and prescribe nutritional interventions, and several approaches have been tried. In this concept paper we review the current evidence on intradialytic nutrition and propose an algorithm for adapting nutritional interventions to individual patients.

Introduction: While the introduction is very informative and well referenced, the presentation of the information could be enhanced by eliminating unnecessary verbiage. 

The rest of the text presented in a logical manner, although some of the terminology used is idiosyncratic. 

Answer.

We tried to shorten the text, and asked for a second reading by our native language editor. 

Figures: The manuscript contains five figures.

  Figures 1 and 3 do not add significant information and can be removed. 

  Figure 5 could be presented as an appendix at the end of the manuscript.

We removed figure 3 and presented figure 5 (now 4) in the appendix. However, we suggest to leave figure 1, due to the crucial importance of vascular disease as a cause of malnutrition. 

Tables: 

  Tables 1 and 2 are well presented summary. Minor criticism: Salt and water restriction, "limits weight gain, improves hypertension and tolerance". It is unclear what tolerance means. 

dialysis tolerance: this was corrected. 

   Tables 3 and 4: I do not believe that these tables are essential to the manuscript as most of the information has been included in the text.    

you are right, but we still think that they may be a useful resume. We would suggest keeping them for this reason. 

Minor criticisms: 

Ketosteril may not be familiar to many readers and as such, should be briefly described. 

Thanks for the suggestion, we rephrased the paragraph as follows:

10. The “case” of ketoacid and aminoacid supplementation.

Amino acid mixtures have been employed in a number of studies to enrich the protein content of the food dialysis patients are offered; their advantage, if any, over other options is not clear, and logistics is probably the most important factor in preferring this approach (124).

In the pre-dialysis phase, plant based, low or very low protein diets supplemented with a mixture of essential aminoacids and ketoacids are generally associated with a slower progression of CKD, and some data on the chronic use of these supplements in dialysis patients are promising, with respect to the preservation of residual kidney function and urine volume output (125-128). We therefore consider the use of this formulation in dialysis patients with low protein intake, and low albumin levels, and who prefer “pills” to other oral supplements.  

The only formulation presently available in France is Ketosteril, consists in film-coated tablets containing ketoanalogues, in forms of calcium salts: a-ketoanalogue of isoleucine, leucine, phenylalanine, valine, methionine; and the following amino-acids: lysine, threonine, tryptophan, histidine, tyrosine. The usual dose, as complement of low protein diets, is one tablet each 5-10 Kg, according to the level of protein restriction. The dose in dialysis patients is not clearly stated; the addition to intradialytic snacks is now under evaluation. While the drawback is the addition of “more pills” to our patients’ already large tablet intake, the advantage is they are less likely to interfere with appetite, thus possibly better preserving spontaneous food intake.

Once again, thank you for your time and interest, and we hope that the revised version may have answered to your requests and comments,

Sincerely, the authors. 

Reviewer 3 Report

This manuscript provides a detailed summary of the state of knowledge of intradialytic nutrition and hemodialysis prescription. The topics discussed are timely and informative. However, there are a few additional points that merit the attention of the authors.

One issue is the need to assess nutritional status before a dialysis session. What are the available methods, their sensitivity and usefulness for the physician?

Hydration status and malnutritional are concurrent liabilities for the HD patient. What is your opinion on the appropriate methods to assess these problems?

You mention anabolic stimuli. I saw no mention of simple supplementation with creatine monohydrate or HMB?

What recommendations are appropriate to conclude your review?

Author Response

First of all, we would like to thank the reviewer for the time dedicated to improve our study, and for the comments and suggestions.

In the following lines we have tried to answer to the questions and comments, aimed at enriching the information. 

This manuscript provides a detailed summary of the state of knowledge of intradialytic nutrition and hemodialysis prescription. The topics discussed are timely and informative. However, there are a few additional points that merit the attention of the authors.

1. One issue is the need to assess nutritional status before a dialysis session. What are the available methods, their sensitivity and usefulness for the physician?

2. Hydration status and malnutritional are concurrent liabilities for the HD patient. What is your opinion on the appropriate methods to assess these problems?

The reviewer is absolutely right. However, discussing this issue would deserve a specific review. We cited a few good reviews and added a table trying to resume the main advantages and limits of each one (table 5)

3. You mention anabolic stimuli. I saw no mention of simple supplementation with creatine monohydrate or HMB?

You are right; we added a paragraph and some references and on these issues, as follows: 

Another question the review did not address is the use of anabolic drugs, such as steroids, or growth hormone, limiting the discussion to “non-pharmacologic” nutritional interventions. The use of anabolic steroids can be indeed considered in selected cases, at least for short periods, but the potential side effects should limit the prescription of these non-routine interventions to particular cases (165-169).

Another suggestion is the use of β-hydroxy-β-methylbutyrate (HMB), a metabolite of leucine able to counteract protein degradation, increase protein synthesis by inhibition of proteasome and stimulation of mTOR pathway, increase the number of mithocondria, reduce pro-oxidant and inflammatory status.  Inprovement of sarcopenia and preservation of muscle function are potential effects of HMB supplementation (170-172). While studies performed in fragile or sarcopenic elderly people are promising on muscle mass and on muscle strength and function, studies in dialysis patients are lacking. A double blind, placebo-controlled, randomized trial of daily HMB supplementation up to 6 months induced no significant advantage in dialysis patients, but a larger scale trial is needed (172). Intradialytic creatine supplementation has been also proposed as an alternative to improve muscle mass and well being. Once more, there is need for large studies to validate these potentially revolutionary approaches (173-174).

4. What recommendations are appropriate to conclude your review?

Thanks for the remark: we tried to summarize them as follows:

The present concept paper suggests employing a stepwise approach against malnutrition in dialysis, starting from nutritional counseling, trying to increase the intake of healthy food, exploiting the potential of intradialytic snacks, followed by nutritional supplements, choosing between different combination, favoring snacks rich in energy or in proteins, upgrading to intradialytic intravenous nutrition, and leaving albumin infusion as a last resort in patients who are metabolically non reactive and severely malnourished patients. Ideally, albumin and intravenous nutrition should be discontinued after nutritional improvement, with the final aim of downgrading the intervention to oral supplements and eventually to increased quantity of health food.

Employing all the available means in a patient-adapted, and whenever possible patient-friendly way, may help to slow the downwards nutritional and clinical spiral often observed in dialysis patients.  

thank you again for your time and advise,

sincerely, the authors. 

Round 2

Reviewer 2 Report

I have reviewed the revised manuscript and agree with the changes that have been incorporated.